# Evaluating Commonalities Across Medically Unexplained Symptoms

**DOI:** 10.3390/ijerph16050818

**Published:** 2019-03-06

**Authors:** Dan Guo, Maria Kleinstäuber, Malcolm Henry Johnson, Frederick Sundram

**Affiliations:** 1Melbourne School of Psychological Sciences, Faculty of Medicine, Dentistry and Health Sciences, University of Melbourne, Victoria 3010, Australia; summer.guo@student.unimelb.edu.au; 2Department of Psychological Medicine, Dunedin Medical School, University of Otago, Dunedin 9016, New Zealand; maria.kleinstaeuber@otago.ac.nz; 3Department of Psychological Medicine, Faculty of Medical and Health Sciences, University of Auckland, Auckland 1142, New Zealand; mh.johnson@auckland.ac.nz

**Keywords:** medically unexplained physical symptoms, MUPS, MUS, cognitive, emotional, psychological, anxiety, depression, quality of life, medical consultation, culture

## Abstract

This commentary presents commonalities in medically unexplained symptoms (MUS) across multiple organ systems, including symptoms, aetiological mechanisms, comorbidity with mental health disorders, symptom burden and impact on quality of life. Further, treatment outcomes and barriers in the clinician–patient relationship, and cross-cultural experiences are highlighted. This discussion is necessary in aiding an improved understanding and management of MUS due to the interconnectedness underlying MUS presentations across the spectrum of medical specialties.

## 1. Introduction

Medically unexplained symptoms (MUS) refer to a broad spectrum of physical symptoms that cannot or have not been sufficiently explained by organic causes after adequate physical examination and investigations [1,2]. MUS can vary in symptom severity, breadth of clinical specialties, and associated levels of care [3]. The impact of MUS can also range from mild to severe disabilities with associated direct and indirect healthcare costs [4,5]. MUS epidemiological data also vary across healthcare settings, whereby a prevalence of 10%–30% has been reported in primary care settings, and 35%–53% in hospital-based care [6]. There is increasing recognition that the aetiology of MUS is complex and arises from an interaction of numerous possible factors, including life experiences, healthcare beliefs, personality, doctor–patient relationship and physiology [7].

However, clinical research in and management of MUS has traditionally been confined to single organ systems. This approach has advanced our understanding of each symptom within the relevant organ system. For example, the aetiology of irritable bowel syndrome has already been explained with a complex interaction between central and psychosocial processes and gut physiology [8], as a result of advances in syndrome-specific research of pathomechanisms over the past decade. However, it also bears several disadvantages when researchers and clinicians develop, dependent on their clinical speciality, an isolated perspective on single, specific syndromes of MUS. It can contribute towards diagnostic confusion. For example, whereas a patient with fatigue and widespread pain in the absence of known medical causes might be diagnosed with chronic fatigue syndrome or fibromyalgia in an internal medicine or rheumatology setting, the same patient would be diagnosed with a somatic symptom disorder by a psychiatrist. Although several patients with fibromyalgia can be also diagnosed with somatic symptom disorder, some studies (e.g., [9]) show that only a small subset of fibromyalgia patients meet criteria of somatic symptom disorder [10]. Moreover, an isolated syndrome-specific perspective can build up barriers to investigating shared mechanisms underlying MUS across the various medical specialties, which would otherwise enhance our understanding of MUS.

MUS can exist as single or multiple symptoms and when they occur as a cluster they are regarded as functional somatic syndromes (FSS). Considerable symptom overlap has been reported across FSS, such as fibromyalgia, chronic fatigue syndrome and irritable bowel syndrome [11]. This finding favours the adoption of a unifying framework where the concept of MUS can be viewed as a single diagnostic category rather than distinct FSS. Existing frameworks utilise a variety of approaches, including symptom count or involvement of particular organ systems, without consideration of a more holistic approach that considers, on an individual level, genetic, biological, psychological, cultural or social determinants [12], but also interactions with healthcare providers within the healthcare system and cultural factors at a population level. In the sections below, consideration is given to how MUS may arise whilst exploring possible aetiological factors, predictors and consequences of MUS involving a broad framework.

## 2. Aetiological Mechanisms

In addition to research in syndrome-specific symptoms, shared aetiological mechanisms have been investigated. One of the most intensively studied concepts in this context is “central sensitisation”—where chronic pain is perpetuated or regulated by a highly reactive state of the nervous system. The term “central sensitivity syndromes” has been used to describe different MUS conditions that appear to share common pathophysiology [13]. There is evidence for central sensitisation in irritable bowel syndrome, fibromyalgia, myofascial pain syndrome, tension headache, multiple chemical syndrome, migraine, temporomandibular joint dysfunction, restless legs syndrome, and primary dysmenorrhoea. Central sensitivity syndromes embrace the biopsychosocial model that posits many factors to modulate the experience of different MUS by their interaction with central sensitisation mechanisms [14]. For example, in fibromyalgia, fatigue is associated with psychological distress; pain is associated with catastrophising and brain activity in areas associated with emotional, perceptual and attentional aspects. Across organ systems, the complex interaction between social factors, early life stressors, prior personal knowledge/experience/belief of symptoms, peripheral sensory inputs, central sensitisation, cognitive processing, attribution/interpretation of symptoms, focus on symptoms and subsequent illness-related behaviours should be considered [15]. 

## 3. Cognitive Factors

Cognitive factors such as symptom attention and catastrophising attribution processes have been demonstrated to be commonly associated with MUS. These factors contribute to different MUS by perpetuating negative illness perceptions and attentional biases towards physical threats, patients’ frequent checking of their symptoms, interpreting normal sensations in a maladaptive way and heightened expectations for negative consequences associated with normal bodily sensations [16,17]. Negative illness perceptions may lead to higher symptom reports and future disability as reflected by impairment in physical, social, and emotional functioning regardless of the organ system.

Van den Bergh and colleagues introduced a model synthesising findings of MUS research and symptom perception [18]. Based on the assumption that the brain makes sense of bodily states according to previous experiences or expectations (i.e., priors), it is proposed that individuals implicitly use an interoceptive process to decide whether sensations are symptoms. Thus, somatic symptoms are conscious percepts resulting from the brain’s interpretation of bodily information in light of priors, which influences how the body is experienced and; thereby, the symptoms reported. Where priors are mismatched to objective sensory inputs (i.e., there is a portion of physiological input not predicted by priors), prediction errors can result and contribute to the lack of accuracy when deciding whether a sensation is a symptom. This model implies that MUS is linked with poor interoceptive accuracy, where excessive confidence is given to priors (in predicting the presence of symptoms) despite there being mismatch with sensory input. The model contributes toward the argument for a unifying framework by adopting the view of MUS sitting on a point along this continuum, which specifies the divergence of objective and subjective health as a result of symptoms perceived (and the interplaying factors that contribute towards a mismatch). This viewpoint is useful in understanding how symptoms that were initially associated with biological markers could eventually become self-reported symptoms lacking a biological basis. It further removes the categorical boundary between MUS and medically explained symptoms by viewing their difference as being on different ends of the continuum, through which gradual and context-dependent changes can occur. Somatosensory amplification [19], for example, a hypervigilance towards physical changes in the body and a disposition to think and feel about bodily changes in a negative, catastrophising way, could be explained in the context of this new model by Van den Bergh and colleagues [18].

## 4. Comorbidity and Symptom Burden

MUS reveal increased comorbidity rates with psychiatric diagnoses, such as anxiety or depression [20]. There is a significantly higher rate of lifetime generalised anxiety disorder, panic disorder and major depressive disorder in people experiencing MUS [21]. However, the temporal relationship to the development of MUS is unclear as mental health disorders can be both contributors or consequences of MUS. For example, shared pathology has been suggested to contribute to the onset of physical and depressive symptoms due to underlying epigenetic processes, such as early childhood trauma affecting the immune system and thus impacting neuronal plasticity through inflammatory pathways [22,23]. 

Additionally, symptom burden, the subjective view of symptom severity and associated physiological burden, is commonly adversely associated with several psychobehavioural and functional characteristics for patients who experience a variety of symptoms of different aetiologies [24]. Patients with irritable bowel syndrome [25], chronic fatigue syndrome [26] and MUS, alongside functional gastrointestinal disorders [27], commonly experience increased symptom burden with increased healthcare utilisation and sick leave days. The level of symptom-related distress is a recognised predictor of MUS prognosis. A further commonality that occurs across MUS is its impact on quality of life. Sufferers of MUS have a greater likelihood of impairments in health-related quality of life [28], long-term occupational functioning [5], and self-rated health [4].

## 5. Treatment Outcome and the Clinician-Patient Relationship

Previous research on aetiological mechanisms has shown that MUS across organ systems can be attributed to a complex interaction of biological, psychological, and social factors. For this reason, treatment of MUS needs a cohesive and interdisciplinary approach [15]. Existing evidence suggests that people with different MUS can respond equally well to similar therapies [29]. Studies have also reported common barriers between physicians and patients during medical consultations, which emphasises the complexity of the interaction process regardless of the type of MUS. These include communication styles, behaviours during consultation, beliefs about treatment in the primary care setting, the extent of problem-exploration, physicians’ attitudes towards patients, a biomedical disease model approach, physicians’ knowledge of MUS, and level of confidence in treatment [30]. Such barriers contribute to reduced opportunities to explore psychosocial factors [31]. In addition, dissatisfaction with care may contribute to the patient’s continued effort in finding better care and thus maintains their re-presenting behaviour to the healthcare system for unmet healthcare needs [32]. There is also a fine balance between excluding organic causes versus over-investigating or overtreatment, which can potentially contribute to iatrogenic harm, while attempting to maintain a therapeutic alliance [33,34].

## 6. Cultural Considerations

Finally, findings from cross-cultural studies suggest that the application of an isolated syndrome-specific perspective across different cultural settings is limited. Although research has demonstrated that MUS are a phenomenon existing across cultures [35], these rates vary substantially. For example, in a primary care study conducted by the World Health Organization, prevalence rates ranged from 7.6% to 36.8%, with highest rates in South American countries [36]. Patterns of most commonly reported somatic symptoms in different cultural settings show similarities: Different kinds of pain (back, head, arms, joint) or fatigue are under the five most commonly reported somatic symptoms reported in various countries (e.g., European countries, Canada, India, China, and South America) [37,38,39,40,41]. However, analysing patterns of self-reported somatic symptoms can result in cross-cultural deviations. A recently published study [42] demonstrated factors of pain-, gastrointestinal- and fatigue-related symptoms across a German, Dutch, and Chinese sample. However cardiopulmonary-related symptoms were grouped in a different way in the Chinese sample compared to both European samples. These deviations can be partly explained by culture-specific perceptions of health and illness. A typical example of the impact of culture-specific perceptions is the “shenjing shuairuo” syndrome in China, which is considered as an equivalent to the Western diagnostic entity of neurasthenia. Although both syndromes have a significant overlap in their symptoms, “shenjing shuairuo” is explained by a more holistic, biopsychosocial approach compared to neurasthenia [43]. Further factors explaining cross-cultural variations regarding prevalence and patterns of MUS are aspects of the doctor–patient relationship. For example, the nature of the doctor-patient relationship modulates rates of MUS presentation to primary care with the higher rates in Latin America thought to be related to the lack of an ongoing therapeutic relationship with their doctor [44]. However, MUS may be a common expression of distress, with higher rates of somatic symptoms being endorsed in some cultures rather than psychological symptoms [45]. 

For instance, the different rates of reported somatic symptoms are thought to arise due to a rather externally-oriented thinking in East Asian cultures (reduced focus on emotions rather than an inability to experience or express them) as opposed to a tendency for reporting psychological symptoms of depression in Western cultures [46]. Additionally, an association between somatic symptoms and depression within Chinese populations [47], but not anxiety [48], has various possible explanations including stigma and help-seeking behaviour [49]. Findings from a South Korean population have suggested that not only the experience and expression of distress but also the conceptualisation and communication of distress are important [50]. In a study of Turkish immigrant women living in Stockholm, illness meaning and somatisation tendencies were examined. The most common symptom reported by the participants was pain. Furthermore, psychiatric attribution and its relevance to recovery as well as one’s own ability to influence his or her recovery were found to be low, while relationships with family and their clinician were regarded as important [51]. Overall, from previous transcultural research on MUS, it can be concluded that somatic symptoms and syndromes of MUS can vary in their prevalence and patterns between countries. However, it is less the symptom or syndrome itself but rather the clinical presentation and the associated features, such as experience and manifestation of distress, which needs to be considered through a cultural lens.

## 7. Conclusions

The aetiology of MUS is complex and often multi-faceted, but a holistic approach towards MUS is not often applied. This paper has highlighted a number of possible factors which contribute to the evolution and perpetuation of MUS. Investigating and managing MUS as confined to single organ systems has enabled a better understanding within each clinical specialty; however, consideration of MUS as a spectrum and at individual, healthcare system and population levels can enhance our understanding of possible shared mechanisms underlying MUS. To date, limited research has been conducted despite sufficient evidence suggesting possible interconnectedness. Furthermore, an integrated approach might add towards improved MUS care, with implications to reducing the perpetuation of illness, disability, and healthcare utilisation. Future research avenues for consideration include the further exploration of the cultural underpinnings of and prevention strategies for MUS, such as correcting misinterpretations of somatic sensations and hypervigilance of interoceptive processes, modulators and consequences of the doctor–patient relationship and cross-cultural screening tools for MUS to facilitate earlier detection and treatment.

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
