# Peer review of "Evaluating Commonalities Across Medically Unexplained Symptoms"

_ijerph, 2019, doi:10.3390/ijerph16050818_

Round 1

Reviewer 1 Report

This commentary aims at showing the commonalities in medically unexplained symptoms (MUS) across different bodily systems. In the past, research and management of MUS were typically confined to one organ system. However, investigating MUS across domains can improve our understanding of mechanisms underlying MUS. Given the high prevalence of MUS, it is important to raise the awareness of this problem to a broader audience.

This article summarizes the recent findings related to MUS and refers to relevant articles and models. This review could be helpful for researchers with no prior experience with MUS. However, in some parts of the paper, the main aim of the paper, i.e., evaluating the commonalities in MUS across different bodily symptoms, is not addressed. Also, the authors could be clearer about which unifying framework they have in mind, as this would be helpful for researchers interested in investigating MUS. Following additional comments are intended to further improve the manuscript.

Line 40-41. It is possible that some patients with fibromyalgia can be also diagnosed with somatic symptom disorder, but some studies (e.g., Häuser et al., 2015) show that only small subset of FM patients met criteria of DSM-5 somatic symptom disorder.

Model of Van den Bergh et al (2017) – The authors could describe this model in more depth and mention which errors can affect the described processes. Also, they could explain how this model fits with their main goal of showing commonalities across MUS.

Comorbidity and symptom burden – Given the aim of the study, I would expect to see information on how comorbidities and burden are similar across MUS in different systems.

Lines 108-109. This sentence is not really clear. I do not understand what is meant with “perpetuates MUS re-presentation if patients hope to seek better care”.

Cultural considerations:

As in “Comorbidity and symptom burden”, the similarities across different MUS are not described.

Adding more information about the prevalence rates in different cultures (How does the prevalence vary? Which MUS in which cultures are the most prevalent?) and the proposed factors is necessary to improve the readability. Right now, this section reads like a description of a few studies without a clear thread.

Lines 124-25 – In what sense important?

Typo: Line 41: by a psychiatrist

References:

Häuser, W., Bialas, P., Welsch, K., & Wolfe, F. (2015). Construct validity and clinical utility of current research criteria of DSM-5 somatic symptom disorder diagnosis in patients with fibromyalgia syndrome. Journal of Psychosomatic Research, 78, 546-552

Author Response

Reviewer 1

This article summarizes the   recent findings related to MUS and refers to relevant articles and models.   This review could be helpful for researchers with no prior experience with   MUS. However, in some parts of the paper, the main aim of the paper, i.e.,   evaluating the commonalities in MUS across different bodily symptoms, is not   addressed. Also, the authors could be clearer about which unifying framework   they have in mind, as this would be helpful for researchers interested in   investigating MUS. Following additional comments are intended to further   improve the manuscript.

The comments from the reviewer   will be discussed in more detail in the relevant sections below.

#R1-01: Line 40-41. It is   possible that some patients with fibromyalgia can be also diagnosed with   somatic symptom disorder, but some studies (e.g., Häuser et al., 2015) show   that only small subset of FM patients met criteria of DSM-5 somatic symptom   disorder.

We thank the reviewer for their   suggestion of this additional reference and have now included this reference   and adapted the reviewer’s point above on fibromyalgia and somatic symptom   disorder. Paragraph 2 has now incorporated the possibility that some patients   with fibromyalgia can be diagnosed with somatic symptom disorder, whilst   other studies (e.g., Häuser et al., 2015) show that only small subset of FM   patients met criteria of DSM-5 somatic symptom disorder (see p. 2).

#R1-02: Model of Van den Bergh   et al (2017) – The authors could describe this model in more depth and   mention which errors can affect the described processes. Also, they could   explain how this model fits with their main goal of showing commonalities   across MUS.

The model of Van den Bergh et   al (2017) has now been described with clearer depth of the model processes   and specific ‘prediction’ errors which can affect these processes. An   additional explanation of how this model would fit with the main goal of   showing commonalities across MUS has also been added (see p. 3).

#R1-03: Comorbidity and symptom   burden – Given the aim of the study, I would expect to see information on how   comorbidities and burden are similar across MUS in different systems.

Considering the aim of this   commentary, further elaboration has been made to provide information on what   comorbidities were observed for which syndromes, as well as how symptom   burden has been similarly associated with different psychobehavioural and   functional characteristics across MUS in different systems (see p. 3).

#R1-04: Lines 108-109. This   sentence is not really clear. I do not understand what is meant with   “perpetuates MUS re-presentation if patients hope to seek better care”.

The sentence “perpetuates MUS   re-presentation if patients hope to seek better care” has now been rephrased   to reflect more clarity around why dissatisfaction can maintain a patient’s   continued re-representing behaviour to healthcare systems (see p. 4).

#R1-05: Cultural   considerations:

As in “Comorbidity and symptom   burden”, the similarities across different MUS are not described.

We thank the reviewer for this   important comment. The first paragraph of the section “Cultural   considerations” focuses now more on cross-cultural similarities in MUS. Study   findings are reported that demonstrate increased prevalence rates of MUS in   many countries all over the world and that demonstrate similarities in most   commonly reported symptoms (see p. 4).

#R1-06: Adding more information   about the prevalence rates in different cultures (How does the prevalence   vary? Which MUS in which cultures are the most prevalent?) and the proposed   factors is necessary to improve the readability. Right now, this section reads   like a description of a few studies without a clear thread.

We followed the reviewer’s   comment and added a study conducted by the WHO that examined prevalence rates   of a syndrome of at least 4 MUS (females)/6 MUS (males) across 14 different   countries. Prevalence rates ranged from 7.6 to 36.8%, with highest rates in   Southern American countries. Although this study is a little bit older, to   our knowledge there is no other more recently published study that examined   MUS across 14 different nations (see p. 4).

Moreover, we changed the   structure of the section “cultural considerations” in order to improve its   readability. We start with cross-cultural similarities, continue with   deviations in prevalence rates and symptom patterns of MUS between countries   and end up with some explanations of these deviations (see p. 4).

#R1-07: Lines 124-25 – In what   sense important?

We split this sentence in two   sentences and reworded it (see p. 5).

#R1-08: Typo: Line 41: by a   psychiatrist

We replaced “from” with “by”   (see p. 2).

Reviewer 2 Report

The article entitled “Evaluating commonalities across medically unexplained symptoms” by Guo et al. is a brief research commentary on commonalities in medically unexplained symptoms (MUS) across multiple organ systems. The paper an interesting presentation and discussion about the different factors involved in the development of medically unexplained symptoms divided in macro-categories.

Although the paper is not a literature review article, the authors should provide some anchors to the existing literature to point out how current research is ignoring the phenomenon.

In this regard, authors can see and add to the references list this recently published article on the subject:

Poloni N., Ielmini M., Caselli I., Ceccon F., Bianchi L., Isella C., Callegari C., Medically unexplained physical symptoms in hospitalized patients: a 9-year retrospective observational study, Frontiers in Psychiatry, 2018, 9(626):1-6.

Moreover, an article of this kind could be educational. I suggest the authors to extend the perspective about how and why attention to this phenomenon is warranted, what are the interesting research questions, and how researchers should develop engage on these questions.

In the conclusions, authors could specify better why future researchers can benefit from ideas in the paper.

Author Response

Reviewer 2

The article entitled   “Evaluating commonalities across medically unexplained symptoms” by Guo et   al. is a brief research commentary on commonalities in medically   unexplained symptoms (MUS) across multiple organ systems. The paper an   interesting presentation and discussion about the different factors involved   in the development of medically unexplained symptoms divided in   macro-categories.

#R2-01: Although the paper is   not a literature review article, the authors should provide some anchors to   the existing literature to point out how current research is ignoring the   phenomenon.

Taking on board the reviewer’s   suggestion on tightening the focus and clarity around commonalities, we have   added key sentences to our Introduction section covering the complexity of   MUS aetiology and possible interacting factors and we also set the scene  better to consider aetiological and predictive factors and consequences of   MUS. We have also included further references from the literature where   relevant throughout the paper. We have expanded the Cognitive factors section   and discussed the Van den Bergh model in more detail (see p. 1-3).

In this regard, authors can see and add to the references list this   recently published article on the subject:

Poloni N., Ielmini M., Caselli   I., Ceccon F., Bianchi L., Isella C., Callegari C., Medically unexplained   physical symptoms in hospitalized patients: a 9-year retrospective   observational study, Frontiers in Psychiatry, 2018, 9(626):1-6.

We thank the reviewer for this   additional reference which is now included in the Introduction section and   which covers several facets that could contribute or be associated with MUS   (see p. 1).

#R2-02: Moreover, an article of   this kind could be educational. I suggest the authors to extend the   perspective about how and why attention to this phenomenon is warranted, what   are the interesting research questions, and how researchers should develop   engage on these questions.

We appreciate this suggestion from the reviewer   and we have attempted to change some of the focus of the paper towards being   an educational piece and we have also added in our Conclusion section   relevant future directions based on our reading of the literature.

From an educational standpoint, the conclusion is   now shaped to reflect how investigating possible shared mechanisms underlying   MUS across the various medical specialties could add towards improved MUS   care, with implications to reducing the perpetuation of illness, disability,   and healthcare utilisation. Hence, attention to the interconnected phenomenon   is warranted. Interesting research questions and ways to engage with them has   also been stated as ‘future research avenues’, including further exploration of   MUS prevention through understanding cultural underpinnings, alteration of   symptom misinterpretation, modulators and consequences of the doctor-patient   relationship, and cross-cultural screening tools for MUS to facilitate   earlier detection and treatment (see p. 5).

#R2-03: In the conclusions, authors could specify better why future   researchers can benefit from ideas in the paper.

This comment from the reviewer is related to the   point above and we would like to draw attention to our updated Conclusion   section which we believe clarifies these aspects better now. We reflect   specific ideas future researchers can benefit from studying, including life   experiences, healthcare beliefs, personality, doctor-patient relationship and   physiological factors (described in this commentary) which contribute to the   evolution and perpetuation of MUS (see p. 1 and 5).

Round 2

Reviewer 1 Report

The authors addressed all my concerns and greatly improved the quality and readability of the manuscript. I have no further comments. Congratulations on your interesting commentary!